# Elevated Serum Protein Induced by Vitamin K Absence or Antagonist II Levels in Patients with Hepatic Hemangiomas

**DOI:** 10.3390/ijms26083681

**Published:** 2025-04-13

**Authors:** Shigeo Maruyama, Tomomitsu Matono, Masahiko Koda

**Affiliations:** 1Maruyama Medical Clinic, Aioi-cho 3921, Hamada 697-0034, Shimane, Japan; 2Department of Gastroenterology, Hyogo Prefectural Harima-Himeji General Hospital, Kamiya-cho 364, Himeji 670-8560, Hyogo, Japan; tonox1976@gmail.com; 3Hino Hospital, Nota 332, Hino 689-4504, Tottori, Japan

**Keywords:** PIVKA-II, hepatic hemangioma, coagulation disorders, prothrombin

## Abstract

Little is known about the effect of hepatic hemangiomas on protein induced by vitamin K absence or antagonist II (PIVKA-II). The aim of this study was to clarify the correlation of PIVKA-II levels with hepatic hemangiomas. In 335 consecutive patients with hepatic hemangiomas, ultrasonography (US), laboratory tests for liver function, serum levels of PIVKA-II and α-fetoprotein (AFP), and coagulation factors (platelets, prothrombin time (PT), fibrinogen, thrombin–antithrombin III complex (TAT), D-dimer, and fibrin and fibrinogen degradation products (FDPs)) as indicators of coagulation disorders were examined. PIVKA-II levels were significantly higher in the hemangioma group than in the control group (*p* < 0.0001), and significantly higher in the large hemangioma group (*p* < 0.0001). PIVKA-II levels in the hemangioma increase group were higher with increases in tumor size and abnormal coagulation factors, and those in the hemangioma decrease group were lower with decreases in tumor size and abnormal coagulation factors. PIVKA-II levels were significantly correlated with tumor size (*p* < 0.0001) and all coagulation factors (*p* < 0.05) except prothrombin. Hepatic hemangiomas were associated with elevated serum PIVKA-II levels, showing significant correlations with tumor size and coagulation disorders. PIVKA-II elevation was attributed to the increased production of prothrombin precursors caused by accelerated coagulation–fibrinolysis within hemangiomas.

## 1. Introduction

Serum protein induced by vitamin K absence or antagonist II (PIVKA-II) levels are often elevated in patients with hepatocellular carcinoma (HCC). PIVKA-II is a sensitive and specific tumor marker for the diagnosis of HCC [1,2], although several other factors such as an insufficiency of vitamin K, administration of vitamin K antagonists, use of antibiotics that alter gut flora, alcoholic liver disease [2], hepatitis E [3], and other liver diseases [4,5] or malignant tumors [6,7] have been reported to increase serum PIVKA-II levels in patients without HCC. However, there have been no previous reports on the association between PIVKA-II levels and hemangiomas. Our previous study demonstrated the relationship between the size of hemangiomas and coagulation factors, and it showed that serum PIVKA-II levels were significantly increased in patients with larger hemangiomas and coagulation disorders [8,9,10]. Therefore, a retrospective study was performed to evaluate the effect of hemangiomas on serum PIVKA-II levels in hemangioma patients without HCC and elucidate the mechanism for the elevation of PIVKA-II levels by hemangiomas.

## 2. Results

### 2.1. Characteristics of Patients and Hemangiomas

The laboratory findings of 335 patients with hepatic hemangiomas are listed in Table 1. There were 122 men and 213 women (male/female ratio, 1:1.7), with a median age of 54 years (range, 21–89 years). The median size of hemangiomas was 20.2 mm (range, 5.1–107 mm). All patients, except one patient with abdominal distension, were asymptomatic, and their hemangiomas had been discovered at a routine health examination or as an incidental finding on radiological studies. At the time of diagnosis, serum laboratory tests were normal in 301 of 335 (89.9%) patients. Thirty-four patients had slight elevations in alanine aminotransferase (ALT), γ-glutamyl transpeptidase (GGT), or alkaline phosphatase (ALP). Fifty-four patients (16.1%) had underlying chronic liver disease. No patient had undergone surgical treatment.

### 2.2. Comparison of Patients and Control Subjects

Table 2 shows a comparison of clinical parameters in 50 control subjects and 335 patients with hepatic hemangiomas. The mean age and sex ratio were similar in the two groups. Albumin levels were significantly lower (*p* < 0.01), and GGT and ALP levels were significantly higher in the patient group than in the control group (*p* < 0.01 and *p* < 0.05, respectively). PIVKA-II and M2BPGi levels were significantly higher in the patient group than in the control group (*p* < 0.0001 and *p* < 0.001, respectively), whereas AFP levels were similar in the two groups. In the patient group of 122 men and 213 women, PIVKA-II levels were significantly higher in women than in men (*p* < 0.01). Platelet counts and PT and fibrinogen levels were similar in the two groups, but TAT, D-dimer, and FDP concentrations were significantly higher in the patient group than in the control group (all *p* < 0.0001). Portal vein diameter and spleen index were significantly elevated in the patient group (both *p* < 0.0001).

### 2.3. Associations Among Hemangioma Size and Clinical Parameters Including Coagulation Factors

Table 3 shows the associations between tumor size and clinical parameters in 335 patients with hepatic hemangiomas. The mean age was significantly older in the large group than in the other groups (*p* < 0.05), but the sex ratio was similar among the four groups. PIVKA-II and M2BPGi concentrations were significantly elevated in the large group (both *p* < 0.0001), whereas AFP levels were not significantly different among the four groups. Platelet counts and fibrinogen levels were significantly lower in the large group than in the other groups (*p* < 0.0001 and *p* < 0.01, respectively). PT levels were similar among the four groups, but TAT, D-dimer, and FDP levels were significantly elevated in the large group (all *p* < 0.0001). Albumin concentration was significantly lower in the large group than in the other groups (*p* < 0.001). Portal vein diameter and spleen index were significantly elevated in the large group (both *p* < 0.0001).

### 2.4. Comparison of Clinical Parameters at the First Examination with Those at the Last Examination in the Follow-Up Period by Changes in Hemangioma Size

Table 4 shows the comparative values between the first and last examinations in each group in terms of changes in size in 232 patients with hepatic hemangiomas. The median follow-up period was 68.6 months in the increase group, 69.7 months in the no-change group, and 76.1 months in the decrease group. The median follow-up period was significantly higher in the decrease group compared to the other groups (*p* < 0.01). In the increase group, PIVKA-II levels were significantly higher at the last examination than at the first examination (*p* < 0.001). Platelet counts (*p* < 0.01) and fibrinogen levels (*p* < 0.05) were significantly lower, and TAT (*p* < 0.01), D-dimer (*p* < 0.001), and FDP levels (*p* < 0.01) were significantly higher at the last examination than at the first examination. In the no-change group, values of PIVKA-II and all coagulation markers were not significantly different between the first and last examinations. In the decrease group, PIVKA-II levels were significantly lower (*p* < 0.001), fibrinogen levels were significantly higher (<0.01), and TAT, D-dimer, and FDP levels were significantly lower (all *p* < 0.001) at the last examination than at the first examination. In all three groups, PT levels were not significantly different between the first and last examinations.

### 2.5. Comparison of Clinical Parameters in Patients with and Without Chronic Liver Disease as an Underlying Disease

Table 5 shows a comparison of clinical parameters in patients with and without chronic liver disease as an underlying disease. Fifty-four patients had chronic liver disease (details in Table 1). PIVKA-II and AFP levels were similar in the two groups, but M2BPGi levels were significantly higher in patients with chronic liver disease than in those without (*p* < 0.01). Platelet counts and fibrinogen levels were significantly lower in patients with chronic liver disease than in those without (*p* < 0.01 and *p* < 0.05, respectively), but PT, TAT, D-dimer, and FDP levels were similar in the two groups. ALT and GGT concentrations were significantly higher in patients with chronic liver disease than in those without (both *p* < 0.001). No effect of liver disease on serum PIVKA-II and PT levels in patients with hemangiomas was observed, as shown in the table.

### 2.6. Correlations Between PIVKA-II and Clinical Parameters

PIVKA-II showed significant correlations with tumor size (*p* < 0.0001), platelet counts (*p* < 0.001), fibrinogen levels (*p* < 0.05), and values of TAT, D-dimer, and FDP (all *p* < 0.0001), but not with PT or AFP (Table 6). Furthermore, a significant correlation was observed between each of the five coagulation factors (i.e., platelet, fibrinogen, TAT, D-dimer, and FDP) (all *p* < 0.05); whereas PT levels were not significantly correlated with any of these five coagulation factors.

## 3. Discussion

PIVKA-II has mostly been used as an indicator of blood coagulation abnormality or vitamin K deficiency, but since Liebman et al. reported in 1984 that PIVKA-II was frequently detected in the serum of patients with HCC [11], PIVKA-II has been widely used as a valuable biomarker for the diagnosis of HCC [1,2]. However, studies on the mechanism causing the elevation in serum PIVKA-II levels in HCC patients have been insufficient. The main possible mechanisms for the production of PIVKA-II by HCC are proposed as follows: (1) increased production of prothrombin precursors [12,13,14,15,16]; (2) decreased activity of γ-glutamyl carboxylase [17,18]; (3) insufficiency of vitamin K [19,20,21]; and (4) a combination of two or more of these factors [1,22,23], although the exact mechanisms still remain to be elucidated.

PIVKA-II, also known as des-γ-carboxy prothrombin, is an abnormal form of prothrombin in which some or all ten γ-carboxyglutamic acid (Gla) residues still remain as glutamic acid (Glu) residues after the incomplete γ-carboxylation of prothrombin precursors [14,16]. Prothrombin, which contains 10 Gla residues in its NH_2_-terminal domain, is primarily synthesized in the liver and is a vitamin K-dependent coagulation factor. Gla residues on prothrombin are synthesized from Glu residues on prothrombin precursors by the vitamin K-dependent enzymatic reaction of γ-glutamyl carboxylase [1,2,12,24]. Vitamin K is an essential cofactor required for the posttranslational carboxylation of Glu residues in prothrombin precursors to form Gla residues. In vitamin K deficiency, 10 Glu residues of prothrombin precursors are not fully carboxylated to Gla residues [12,15,25,26], and incompletely carboxylated prothrombin is released into the blood as PIVKA-II.

Hepatic hemangiomas are the most common benign tumors of the liver. Most hemangiomas are small and asymptomatic; therefore, these tumors have little clinical significance. In view of this, little is known of the role of tumor markers, such as PIVKA-II, in hemangiomas, and there have been no previous reports on the relationship between PIVKA-II and hemangiomas. We previously demonstrated that differences in the size of hemangiomas were caused by intratumoral thrombosis and subsequent hemorrhage [8,9,10], and we further found that PIVKA-II levels were significantly elevated in patients with larger hemangiomas and coagulation disorders [9], suggesting that hemangiomas were associated with an elevation in serum PIVKA-II levels, with significant correlations with tumor size and coagulation disorders.

The present study demonstrated that platelet counts and fibrinogen levels were significantly decreased, and TAT, D-dimer, and FDP levels were significantly elevated along with an increase in hemangioma size by an accelerated coagulation–fibrinolysis triggered by intratumoral thrombosis. Under the above conditions, PT levels should theoretically be reduced by the consumption of prothrombin, but the present results showed that PT did not differ significantly among patients with various types of tumors, and abnormal PT values were found in only 2 (0.6%) of 335 patients. Furthermore, PT values were not significantly correlated with TAT levels. TAT is generated in the process of the conversion of prothrombin to thrombin. Therefore, values of prothrombin generally decrease with an increase in TAT levels. The exact reason why this discrepancy between values of prothrombin and TAT occurred in the present study is unknown, but it is suggested that a normal level of serum prothrombin is considered a result of a compensatory increased production of prothrombin precursors in response to the consumption of prothrombin caused by an accelerated coagulation–fibrinolysis. Thus, it follows that the synthesis of prothrombin increased to compensate for the consumed amounts of prothrombin.

Based on our results, we speculated on the possible relationship between each coagulation factor in an accelerated coagulation–fibrinolysis system in hepatic hemangiomas (Figure 1). A previous study reported that fibrinogen levels were correlated with platelet counts [27], and our results also demonstrated that fibrinogen levels were significantly correlated with platelet counts. Low platelet counts and fibrinogen levels were considered the result of intravascular coagulation in hemangiomas [28]. Almost all patients with elevated levels of D-dimer and FDP exhibited increased levels of TAT. Furthermore, a significant correlation was observed between each of the five coagulation factors (platelet, fibrinogen, TAT, D-dimer, and FDP) except prothrombin. Therefore, it is assumed that prothrombin behaves differently from these five other coagulation factors in the process of the coagulation–fibrinolysis, resulting from a compensatory increased synthesis of prothrombin in hepatic cells.

The present results relating to the change in PIVKA-II levels by tumor growth demonstrated an elevation in PIVKA-II levels in patients with increases in tumor size and abnormal coagulation factors, no change in patients with stable lesions, and a decrease in patients with decreases in tumor size and abnormal coagulation factors. Furthermore, PIVKA-II levels were significantly correlated with tumor size and all coagulation factors except prothrombin. From the above results, PIVKA-II levels were closely associated with changes in hemangioma size and abnormal coagulation factors.

PIVKA-II levels exceeding the 40 mAU/mL cutoff value were found in six patients (1.8%), whose tumors were ruled out as HCC by the follow-up studies of changes in findings on US and CT, and of PIVKA-II and AFP levels. The characteristic findings of these six patients were as follows: (1) mean PIVKA-II of 65 mAU/mL (range, 52–88 mAU/mL); (2) mean tumor size of 51.3 mm (range, 40.5–63.4 mm); (3) abnormally low platelet counts and fibrinogen levels; (4) abnormally high TAT, D-dimer, and FDP levels; and (5) PT values were all normal. In the follow-up study of the above 6 patients, serum PIVKA-II levels in 5 patients decreased along with decreases in tumor size and abnormal coagulation factors, and those in the remaining 1 patient were elevated, with increases in tumor size and abnormal coagulation factors. These findings indicate that high serum PIVKA-II levels were more frequent in patients with larger tumors and coagulation disorders, and their levels were closely related to the changes in hemangioma size and coagulation disorders.

Based on these research results, we propose a mechanism for the production of PIVKA-II by hepatic hemangiomas (Figure 1). Prothrombin levels decrease with the consumption of prothrombin by the accelerated coagulation–fibrinolysis within hemangiomas that results in a compensatory increased production of prothrombin precursors in hepatic cells, which leads to increased demand for vitamin K or γ-glutamyl carboxylase, and causes a deficiency in vitamin K and a subsequent decrease in γ-glutamyl carboxylase activity. As a result, some excessively synthesized prothrombin precursors cannot be fully carboxylated, incompletely carboxylated prothrombin appears in plasma as PIVKA-II, and other precursors are completely carboxylated and secreted into the blood as normal prothrombin; consequently, the serum level of PIVKA-II is elevated, and prothrombin levels return to normal. From the above, it is concluded that the elevation in serum PIVKA-II levels in patients with hemangiomas is attributed to an increased production of prothrombin precursors caused by an accelerated coagulation–fibrinolysis within hemangiomas.

The present study has several limitations. First, serum levels of prothrombin precursors were not measured, and thus the increased formation of prothrombin precursors was not serologically proven; it has only been indirectly proven according to the trends in prothrombin levels. Second, the levels of prothrombin, vitamin K, and PIVKA-II in liver tissues were not measured, and our views were established based on the serological levels; therefore, measurements and results at the histological level are needed to validate the role of PIVKA-II in hemangiomas. Third, the length of follow-up and the number of patients, especially those with giant hemangiomas, were limited; therefore, a longer follow-up period and a larger patient cohort with giant hemangiomas are needed to evaluate the elevation in serum PIVKA-II levels. Therefore, further studies are required to evaluate the exact relationship between hemangiomas and the elevation in serum PIVKA-II levels.

## 4. Materials and Methods

### 4.1. Patients

This study was approved by the ethics review board of Tottori University Hospital (approval number: 18A023) and the ethics committee of Hino Hospital (approval number: 2018-4). Of 27,171 abdominal ultrasonography (US) examinations performed at our hospital, Hino Hospital, and Tottori University Hospital, Japan, between January 2016 and May 2024, 335 patients were diagnosed with hepatic hemangiomas and were consecutively enrolled in this study after giving their informed consent. Of the 335 patients initially diagnosed with hepatic hemangiomas, 232 patients with a follow-up period of at least four years were retrospectively enrolled in this follow-up study. The median follow-up period was 71.8 months (range, 48–99 months). Patients with infectious diseases, liver cirrhosis, or malignant tumors were excluded from the study. Patients with abnormal coagulation factors were referred for cardiovascular medicine consultation to prevent complications of vascular thrombotic diseases. No patients were treated with vitamin K or vitamin K antagonist and anti-coagulation therapy. Fifty control subjects were categorized and matched by age, sex, and absence of liver disease.

### 4.2. Methods

Hepatic hemangiomas were diagnosed by US and multiphase contrast-enhanced helical computed tomography (CT). Using the categorization described in a previous study [29], hepatic hemangiomas were divided into three groups according to their maximum diameter: small group (<20 mm); medium group (20–40 mm); and large group (>40 mm). Follow-up was performed with repeated US examinations and laboratory tests for liver function, including PIVKA-II and α-fetoprotein (AFP), and coagulation factors (platelets, prothrombin time (PT), fibrinogen, TAT, D-dimer, and FDP) every 3–6 months at the time of a regular medical examination. The standard rate of change in tumor size was established, and the changes in size were classified into three groups according to the rate of change: (1) decrease, change ≤ −10%; (2) no change, change > −10% to <+10%; and (3) increase, change ≥ +10%. Changes in tumor size within an upper or lower limit of 2 mm were not considered important, due to the measurement limitations of US [30].

Routine laboratory tests were performed using automated methods. Blood platelet counts in the range of 14–38 × 10^4^/μL were considered normal. PT was determined by the coagulation method using a one-stage prothrombin-time assay, and the percentage of prothrombin activity (PT (%)) was adopted as the notation for PT, with values in the range of 70–130% considered normal. Fibrinogen was determined by a rapid physiological coagulation technique using the clotting method, with values in the range of 180–400 mg/dL considered normal. The plasma TAT concentration was determined by a chemiluminescent enzyme immunoassay, with a TAT level of <3.0 ng/mL considered normal. D-dimer and FDP concentrations were determined by a latex immunoturbidimetric assay. Normal D-dimer and FDP concentrations are <1.0 μg/mL and <5.0 μg/mL, respectively. PIVKA-II and AFP concentrations were determined by a chemiluminescent enzyme immunoassay, and normal values of PIVKA-II and AFP are <40 mAU/mL and ≤10 ng/mL, respectively [31,32].

### 4.3. Statistical Analysis

All measurements are expressed as the mean ± standard deviation (SD). Significant differences among two or more groups were determined using Student’s *t*-test or one-way analysis of variance with the Kruskal–Wallis test and Dunn’s test as a post hoc test. Categorical variables were analyzed using the Chi-squared test. The Bonferroni method was used as the post hoc test. The differences between clinical parameters at the first examination and the last examination were assessed using a paired *t*-test. Correlation analysis was carried out by univariate linear regression analysis. The data were analyzed using Stat Flex version 6.0 (Artech Co., Ltd., Osaka, Japan). A *p*-value of <0.05 was considered significant.

## 5. Conclusions

The present study is the first to demonstrate the correlation of hepatic hemangiomas with serum PIVKA-II levels. Hemangiomas were significantly associated with elevated serum PIVKA-II levels, which was attributed to an increased production of prothrombin precursors caused by the accelerated coagulation–fibrinolysis within hemangiomas. Although further studies are required to evaluate the exact relationship between hepatic hemangiomas and PIVKA-II levels, we recommend that elevated serum PIVKA-II levels in patients with hemangiomas should be interpreted with close attention to differentiating them from HCC.

## Figures and Tables

**Figure 1 ijms-26-03681-f001:**
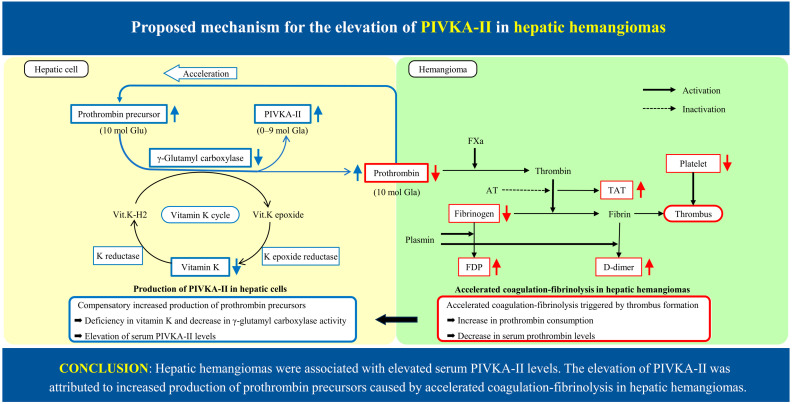
The production of PIVKA-II and normal prothrombin in hepatic cells and accelerated coagulation–fibrinolysis in hepatic hemangioma. Prothrombin levels decrease with the consumption of prothrombin by the accelerated coagulation–fibrinolysis within hepatic hemangiomas. This results in a compensatory increased production of prothrombin precursors in hepatic cells, which leads to increased demand for vitamin K or γ-glutamyl carboxylase and causes a deficiency in vitamin K and a subsequent decrease in γ-glutamyl carboxylase activity. As a result, some excessively synthesized prothrombin precursors cannot be fully carboxylated, and incompletely carboxylated prothrombin appears in plasma as PIVKA-II. Other precursors are completely carboxylated and secreted into the blood as normal prothrombin. Consequently, the serum level of PIVKA-II is elevated, and prothrombin levels return to normal.

**Table 1 ijms-26-03681-t001:** Laboratory findings of 335 patients with hepatic hemangiomas.

Parameters	Value
Age (years)	54 ± 15
Male/female (*n*)	122/213
Biochemistry	
Total bilirubin (mg/dL)	0.6 ± 0.2
Albumin (g/dL)	4.2 ± 0.2
ALT (U/L)	20 ± 14
GGT (U/L)	38 ± 40
ALP (U/L)	234 ± 74
BUN (mg/dL)	14.2 ± 3.7
Cr (mg/dL)	0.70 ± 0.17
Serology	
PIVKA-II (mAU/mL)	21.6 ± 8.7
AFP (ng/mL)	3.6 ± 1.4
M2BPGi (COI)	0.57 ± 0.32
Hematology	
Hemoglobin (g/dL)	13.6 ± 1.3
Platelet (10^4^/μL)	22.1 ± 5.3
Coagulation	
PT (%)	95.2 ± 12.5
Fibrinogen (mg/dL)	285 ± 73
TAT (ng/mL)	1.42 ± 1.04
D-dimer (μg/mL)	0.68 ± 0.70
FDP (μg/mL)	1.69 ± 1.05
Echo findings	
Size of hemangioma (mm)	20.2 ± 16.3
Small < 20 (*n*)	215 (64.2%)
Medium 20–40 (*n*)	87 (26.0%)
Large > 40 (*n*)	33 (9.8%)
Location of hemangioma (*n*)	
Right lobe	285 (85.1%)
Left lobe	48 (14.3%)
Bilateral lobe	2 (0.6%)
Number of hemangioma (*n*)	
Single	290 (86.6%)
Multiple	45 (13.4%)
Associated liver diseases (*n*)	
Hepatitis B	12 (3.6%)
Hepatitis C	7 (2.1%)
Autoimmune hepatitis	7 (2.1%)
Primary biliary cholangitis	3 (0.9%)
Alcoholic liver disease	16 (4.8%)
Nonalcoholic steatohepatitis	9 (2.7%)

All data represent values at the time of study enrollment. Values are presented as the mean ± standard deviation or number (%). ALT, alanine aminotransferase; GGT, γ-glutamyl transpeptidase; ALP, alkaline phosphatase; BUN, blood urea nitrogen; Cr, creatinine; PIVKA-II, protein induced by vitamin K absence or antagonist-II; AFP, α-fetoprotein; M2BPGi, Mac-2 binding protein glycosylation isomer; PT, prothrombin time; TAT, thrombin–antithrombin III complex; FDP, fibrin and fibrinogen degradation product.

**Table 2 ijms-26-03681-t002:** Comparison of clinical parameters in 50 controls and 335 patients with hepatic hemangiomas.

Parameters	Control (*n* = 50)	Patient (*n* = 335)	*p* Value
Age (years)	56 ± 13	54 ± 15	0.1941
Male/female (*n*)	18/32	122/213	0.6225
Biochemistry			
Total bilirubin (mg/dL)	0.6 ± 0.2	0.6 ± 0.2	0.9903
Albumin (g/dL)	4.3 ± 0.2	4.2 ± 0.2	0.0045
ALT (U/L)	19 ± 8	20 ± 14	0.3531
GGT (U/L)	30 ± 15	38 ± 40	0.0053
ALP (U/L)	216 ± 47	234 ± 74	0.0286
BUN (mg/dL)	13.3 ± 4.0	14.2 ± 3.7	0.1250
Cr (mg/dL)	0.69 ± 0.15	0.70 ± 0.17	0.5640
Serology			
PIVKA-II (mAU/mL)	17.4 ± 2.8	21.6 ± 8.7	<0.0001
AFP (ng/mL)	3.6 ± 1.5	3.6 ± 1.4	0.8921
M2BPGi (COI)	0.45 ± 0.18	0.57 ± 0.32	0.0001
Hematology			
Hemoglobin (g/dL)	13.7 ± 1.2	13.6 ± 1.3	0.6254
Platelet (10^4^/μL)	22.1 ± 4.7	22.1 ± 5.3	0.9903
Coagulation			
PT (%)	96.0 ± 11.1	95.2 ± 12.5	0.6087
Fibrinogen (mg/dL)	298 ± 87	285 ± 73	0.3067
TAT (ng/mL)	1.05 ± 0.48	1.42 ± 1.04	<0.0001
D-dimer (μg/mL)	0.46 ± 0.26	0.68 ± 0.66	<0.0001
FDP (μg/mL)	1.36 ± 0.25	1.69 ± 0.99	<0.0001
Echo findings			
Portal vein diameter (mm)	9.4 ± 1.6	10.7 ± 2.2	<0.0001
Spleen index (mm^2^)	1045 ± 370	1302 ± 537	<0.0001

All data represent values at the time of study enrollment. Values are presented as the mean ± standard deviation. ALT, alanine aminotransferase; GGT, γ-glutamyl transpeptidase; ALP, alkaline phosphatase; BUN, blood urea nitrogen; Cr, creatinine; PIVKA-II, protein induced by vitamin K absence or antagonist-II; AFP, α-fetoprotein; M2BPGi, Mac-2 binding protein glycosylation isomer; PT, prothrombin time; TAT, thrombin–antithrombin III complex; FDP, fibrin and fibrinogen degradation product.

**Table 3 ijms-26-03681-t003:** Association between tumor size and clinical parameters in 50 controls and 335 patients with hepatic hemangiomas.

Parameters	Control (*n* = 50)	Small (*n* = 215)	Medium (*n* = 87)	Large (*n* = 33)	*p* Value
Age (years)	56 ± 13	52 ± 15	55 ± 14	59 ± 13	0.0164
Male/female (*n*)	18/32	72/143	35/52	15/18	0.6782
Serology					
PIVKA-II (mAU/mL)	17.4 ± 2.8	19.1 ± 5.1	22.3 ± 7.7 **^,##^	34.0 ± 9.5 **^,##,$$^	<0.0001
AFP (ng/mL)	3.6 ± 1.5	3.4 ± 1.3	3.8 ± 1.5	3.4 ± 1.7	0.0564
M2BPGi (COI)	0.45 ± 0.18	0.48 ± 0.25	0.62 ± 0.34 **^,##^	0.92 ± 0.35 **^,##,$$^	<0.0001
Hematology					
Hemoglobin (g/dL)	13.7 ± 1.2	13.6 ± 1.3	13.8 ± 1.3	13.1 ± 1.7	0.2114
Platelet (10^4^/μL)	22.1 ± 5.4	23.1 ± 4.9	21.4 ± 5.4 ^#^	17.9 ± 4.1 **^,##,$$^	<0.0001
Coagulation					
PT (%)	96.0 ± 11.1	94.5 ± 12.5	95.6 ± 13.3	95.5 ± 8.7	0.7671
Fibrinogen (mg/dL)	298 ± 87	294 ± 74	275 ± 66	246 ± 67 *^,##^	0.0016
TAT (ng/mL)	1.05 ± 0.48	1.12 ± 0.65	1.43 ± 0.77 ^#^	3.39 ± 1.65 **^,##,$$^	<0.0001
D-dimer (μg/mL)	0.46 ± 0.26	0.46 ± 0.27	0.64 ± 0.36 ^##^	2.19 ± 1.16 **^,##,$$^	<0.0001
FDP (μg/mL)	1.36 ± 0.25	1.36 ± 0.34	1.57 ± 0.58 ^#^	4.09 ± 1.69 **^,##,$$^	<0.0001
Biochemistry					
Total bilirubin (mg/dL)	0.6 ± 0.2	0.6 ± 0.2	0.6 ± 0.3	0.6 ± 0.2	0.9474
Albumin (g/dL)	4.3 ± 0.2	4.2 ± 0.2	4.3 ± 0.2	4.0 ± 0.3 **^,##,$$^	0.0002
ALT (U/L)	19 ± 8	21 ± 16	20 ± 10	16 ± 6	0.7291
GGT (U/L)	30 ± 15	43 ± 59	43 ± 40	30 ± 29	0.1987
ALP (U/L)	216 ± 47	230 ± 73	234 ± 68	236 ± 82	0.4517
Echo findings					
Portal vein diameter (mm)	9.4 ± 1.6	10.1 ± 2.1	11.3 ± 1.7 **	12.8 ± 2.3 **^,##,$$^	<0.0001
Spleen index (mm^2^)	1045 ± 370	1231 ± 499	1326 ± 494	1674 ± 714 **^,##,$^	<0.0001

All data represent values at the time of study enrollment. Values are presented as the mean ± standard deviation. *: *p* < 0.05 compared to controls, **: *p* < 0.01 compared to controls, ^#^: *p* < 0.05 compared to the small group, ^##^: *p* < 0.01 compared to the small group, ^$^: *p* < 0.05 compared to the medium group, ^$$^: *p* < 0.01 compared to the medium group, PIVKA-II, protein induced by vitamin K absence or antagonist-II; AFP, α-fetoprotein; M2BPGi, Mac-2 binding protein glycosylation isomer; PT, prothrombin time; TAT, thrombin–antithrombin III complex; FDP, fibrin and fibrinogen degradation product; ALT, alanine aminotransferase; GGT, γ-glutamyl transpeptidase; ALP, alkaline phosphatase.

**Table 4 ijms-26-03681-t004:** Comparison between the first and last values for the follow-up period among each group of changes in size in 232 patients with hepatic hemangiomas.

	Increase (*n* = 71)			No Change (*n* = 74)			Decrease (*n* = 87)		
Parameters	First	Last	*p* Value	First	Last	*p* Value	First	Last	*p* Value
Tumor size (mm)	18.9 ± 10.4	25.0 ± 13.4	<0.001	14.2 ± 7.9	14.0 ± 8.2	0.220	32.1 ± 22.8	23.6 ± 20.0	<0.001
Serology									
PIVKA-II (mAU/mL)	20.5 ± 6.7	25.9 ± 9.0	<0.001	20.4 ± 5.1	21.1 ± 5.1	0.622	25.9 ± 9.1	21.3 ± 7.0	<0.001
AFP (ng/mL)	3.5 ± 1.5	3.7 ± 1.7	0.297	3.5 ± 1.6	3.8 ± 1.7	0.154	3.6 ± 1.5	3.8 ± 1.8	0.171
M2BPGi (COI)	0.58 ± 0.29	0.51 ± 0.25	0.003	0.49 ± 0.21	0.55 ± 0.27	0.014	0.74 ± 0.37	1.06 ± 0.48	<0.001
Hematology									
Hemoglobin (g/dL)	13.7 ± 1.3	13.7 ± 1.5	0.726	13.8 ± 1.1	13.9 ± 1.1	0.965	13.4 ± 1.3	13.2 ± 1.3	0.115
Platelet (10^4^/μL)	23.4 ± 4.7	22.1 ± 4.7	0.006	22.3 ± 5.4	21.2 ± 4.7	0.021	20.4 ± 5.4	20.6 ± 5.4	0.422
Coagulation									
PT (%)	93.7 ± 11.2	96.5 ± 10.1	0.393	93.3 ± 12.8	91.8 ± 10.1	0.168	96.1 ± 12.5	96.5 ± 10.5	0.487
Fibrinogen (mg/dL)	279 ± 68	267 ± 55	0.015	284 ± 69	273 ± 69	0.175	264 ± 63	281 ± 54	0.001
TAT (ng/mL)	1.32 ± 0.86	1.78 ± 1.14	0.001	1.05 ± 0.50	1.09 ± 0.67	0.669	1.89 ± 1.38	1.38 ± 1.01	<0.001
D-dimer (μg/mL)	0.57 ± 0.35	0.72 ± 0.39	<0.001	0.49 ± 0.33	0.45 ± 0.31	0.411	1.07 ± 0.97	0.69 ± 0.59	<0.001
FDP (μg/mL)	1.49 ± 0.52	1.65 ± 0.69	0.001	1.36 ± 0.38	1.43 ± 0.49	0.254	2.09 ± 1.43	1.69 ± 0.95	<0.001
Biochemistry									
Total bilirubin (mg/dL)	0.6 ± 0.2	0.6 ± 0.2	0.068	0.6 ± 0.3	0.6 ± 0.2	0.624	0.6 ± 0.2	0.6 ± 0.2	0.141
Albumin (g/dL)	4.3 ± 0.2	4.3 ± 0.2	0.092	4.3 ± 0.2	4.3 ± 0.2	0.141	4.1 ± 0.3	4.1 ± 0.3	0.415
ALT (U/L)	19 ± 10	21 ± 10	0.066	19 ± 9	21 ± 10	0.146	22 ± 18	21 ± 10	0.835
GGT (U/L)	46 ± 41	49 ± 45	0.347	33 ± 27	38 ± 32	0.011	45 ± 69	47 ± 58	0.494
ALP (U/L)	215 ± 57	227 ± 57	0.187	233 ± 87	241 ± 85	0.040	244 ± 66	265 ± 65	0.002
Echo findings									
Portal vein diameter (mm)	11.2 ± 1.8	11.2 ± 2.1	0.877	10.4 ± 2.1	10.4 ± 1.9	0.891	11.0 ± 2.2	11.2 ± 2.3	0.298
Spleen index (mm^2^)	1449 ± 549	1430 ± 506	0.715	1151 ± 469	1138 ± 472	0.670	1274 ± 530	1312 ± 518	0.213

Values are presented as the mean ± standard deviation. PIVKA-II, protein induced by vitamin K absence or antagonist-II; AFP, α-fetoprotein; M2BPGi, Mac-2 binding protein glycosylation isomer; PT, prothrombin time; TAT, thrombin–antithrombin III complex; FDP, fibrin and fibrinogen degradation product; ALT, alanine aminotransferase; GGT, γ-glutamyl transpeptidase; ALP, alkaline phosphatase.

**Table 5 ijms-26-03681-t005:** A comparison of clinical parameters in patients with and without chronic liver disease as an underlying disease.

Parameters	Liver Disease (+) *n* = 54	Liver Disease (−) *n* = 281	*p* Value
Age (years)	58 ± 12	53 ± 15	0.0067
Male/female (*n*)	28/26	94/187	0.6220
Serology			
PIVKA-II (mAU/mL)	22.6 ± 7.8	21.4 ± 8.9	0.2676
AFP (ng/mL)	3.5 ± 1.7	3.6 ± 1.4	0.7923
M2BPGi (COI)	0.69 ± 0.34	0.54 ± 0.31	0.0035
Hematology			
Hemoglobin (g/dL)	14.1 ± 1.5	13.5 ± 1.3	0.0161
Platelet (10^4^/μL)	20.3 ± 5.5	22.8 ± 5.1	0.0035
Coagulation			
PT (%)	95.9 ± 13.8	95.1 ± 12.2	0.6980
Fibrinogen (mg/dL)	265 ± 72	288 ± 72	0.0147
TAT (ng/mL)	1.53 ± 1.34	1.40 ± 1.06	0.4532
D-dimer (μg/mL)	0.75 ± 0.67	0.67 ± 0.70	0.3879
FDP (μg/mL)	1.84 ± 1.13	1.66 ± 1.03	0.2548
Biochemistry			
Total bilirubin (mg/dL)	0.6 ± 0.3	0.6 ± 0.2	0.4179
Albumin (g/dL)	4.2 ± 0.3	4.2 ± 0.2	0.3087
ALT (U/L)	31 ± 26	18 ± 8	0.0005
GGT (U/L)	66 ± 67	33 ± 29	0.0005
ALP (U/L)	247 ± 86	231 ± 71	0.2364
BUN (mg/dL)	14.3 ± 3.4	14.2 ± 3.8	0.8496
Cr (mg/dL)	0.75 ± 0.18	0.69 ± 0.17	0.0492
Echo findings			
Tumor size (mm)	23.5 ± 15.6	19.5 ± 16.3	0.0888
Portal vein diameter (mm)	11.6 ± 2.2	10.5 ± 2.2	0.0010
Spleen index (mm^2^)	1385 ± 498	1286 ± 543	0.2048

All data represent values at the time of study enrollment. Values are presented as the mean ± standard deviation. PIVKA-II, protein induced by vitamin K absence or antagonist-II; AFP, α-fetoprotein; M2BPGi, Mac-2 binding protein glycosylation isomer; PT, prothrombin time; TAT, thrombin–antithrombin III complex; FDP, fibrin and fibrinogen degradation products; ALT, alanine aminotransferase; GGT, γ-glutamyl transpeptidase; ALP, alkaline phosphatase; BUN, blood urea nitrogen; Cr, creatinine.

**Table 6 ijms-26-03681-t006:** Correlations between PIVKA-II and clinical parameters in 335 patients with hepatic hemangiomas.

Parameters	r	*p* Value
Tumor size (mm)	0.4629	<0.0001
Platelet (10^4^/μL)	−0.1883	0.0005
PT (%)	0.0408	0.4566
Fibrinogen (mg/dL)	−0.1121	0.0403
TAT (ng/mL)	0.4328	<0.0001
D-dimer (μg/mL)	0.5357	<0.0001
FDP (μg/mL)	0.5303	<0.0001
AFP (ng/mL)	0.0785	0.1518

PT, prothrombin time; TAT, thrombin–antithrombin III complex; FDP, fibrin and fibrinogen degradation product; AFP, α-fetoprotein.

## Data Availability

The data presented in this study are available on request from the corresponding author.

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
