# Peer review of "Elevated Serum Protein Induced by Vitamin K Absence or Antagonist II Levels in Patients with Hepatic Hemangiomas"

_ijms, 2025, doi:10.3390/ijms26083681_

Round 1

Reviewer 1 Report

Comments and Suggestions for Authors

I have the following comments:

1) I am somewhat confused by the statements in the Introduction (lines 40-43): “serum PIVKA-II levels were significantly increased in patients with larger hemangiomas and coagulation disorders [7–9]. There have been no previous reports of the association between PIVKA-II levels and hemangiomas”. The first sentence indicates that a previous study reported an association between PIVKA-II levels and hemangiomas, while the next sentence claims that there are no previous reports about this association. It would be helpful if the authors could clarify the novelties of this study and what distinguishes it from existing studies.

2) Do the authors believe that multiple hypothesis correction should be applied to the statistical tests conducted in this study?

3) Given that the study involves 335 participants, including 122 males and 213 females, have the authors explored any potential difference in PIVKA-II association between males and females?

Author Response

Reviewer 1

1) I am somewhat confused by the statements in the Introduction (lines 40-43): “serum PIVKA-II levels were significantly increased in patients with larger hemangiomas and coagulation disorders [7–9]. There have been no previous reports of the association between PIVKA-II levels and hemangiomas”. The first sentence indicates that a previous study reported an association between PIVKA-II levels and hemangiomas, while the next sentence claims that there are no previous reports about this association. It would be helpful if the authors could clarify the novelties of this study and what distinguishes it from existing studies.

Reply

According to reviewer’s comments, we changed the latter half of introduction in order to clarify the novelties of our study.

   Page 3, Line 8

   Introduction 

     Serum protein induced by vitamin K absence or antagonist II

   (PIVKA-II) levels are often elevated in patients with hepatocellular

   carcinoma (HCC). PIVKA-II is a sensitive and specific tumor marker

   for the diagnosis of HCC [1,2], although several other factors such as

   insufficiency of vitamin K, administration of vitamin K antagonists, use

   of antibiotics that alter gut flora, alcoholic liver disease [2], hepatitis E

   [3], and other liver diseases [4,5] or malignant tumors [6,7] have been

   reported to increase serum PIVKA-II levels in patients without HCC.

   However, there have been no previous reports of the association

   between PIVKA-II levels and hemangiomas. Our previous study

   demonstrated the relationship between the size of hemangiomas and

   coagulation factors, and it showed that serum PIVKA-II levels were

   significantly increased in patients with larger hemangiomas and

   coagulation disorders [8-10]. Therefore, a retrospective study was

   performed to evaluate the effect of hemangiomas on serum PIVKA-II

   levels in hemangioma patients without HCC and elucidate the

   mechanism for the elevation of PIVKA-II levels by hemangiomas.

2) Do the authors believe that multiple hypothesis correction should be applied to the statistical tests conducted in this study?

Reply

We performed a postHoc analysis using the Bonferroni method, and the results are shown in table 3.

We have added following sentence in statistical analysis on page 11, line 11:

   The Bonferroni method was used as the post Hoc test.

3) Given that the study involves 335 participants, including 122 males and 213 females, have the authors explored any potential difference in PIVKA-II association between males and females?

Reply

We used an unpaired t-test to examine sex differences in PIVKA-II. The results showed that PIVKA-II was significantly higher in women than in men.

We have added following sentences on page 4, line 7:

In the patient group of 122 men and 213 women, PIVKA-II levels were significantly higher in women than in men (p<0.01).

Reviewer 2 Report

Comments and Suggestions for Authors

General Considerations

This study addresses a relatively unexplored topic: the correlation between serum PIVKA-II levels and hepatic hemangiomas. The approach is innovative and could have significant clinical implications, particularly in differentiating hepatic hemangiomas from hepatocellular carcinoma (HCC). The study is based on a large cohort (335 patients) with good follow-up, and the statistical analyses (t-test, ANOVA, linear regression) are appropriately applied. Well-structured tables facilitate data interpretation.

However, the paper requires a major revision in the following areas:

  • Introduction: Additional references should be included to highlight the role of PIVKA-II as a biomarker in various neoplasms, such as pancreatic ductal adenocarcinoma (PDAC), emphasizing its emerging relevance beyond HCC.
  • Hypothesis: The proposed compensatory role of prothrombin precursor production is based on indirect evidence. Therefore, data on serum precursor levels should be incorporated to support this hypothesis.
  • Results: Histological analyses confirming PIVKA-II production in tissues should be included.
  • Discussion: Some sections are redundant and should be streamlined. Additionally, a comparison with findings from other studies would be valuable, highlighting similarities, differences, and potential clinical implications.
  • Conclusion: The conclusion should be more impactful, emphasizing the clinical significance and implications of the findings.
  • Limitations: Consider adding a clearer section outlining the study’s limitations.
Comments on the Quality of English Language

The english language should be improved

Author Response

General Considerations

This study addresses a relatively unexplored topic: the correlation between serum PIVKA-II levels and hepatic hemangiomas. The approach is innovative and could have significant clinical implications, particularly in differentiating hepatic hemangiomas from hepatocellular carcinoma (HCC). The study is based on a large cohort (335 patients) with good follow-up, and the statistical analyses (t-test, ANOVA, linear regression) are appropriately applied. Well-structured tables facilitate data interpretation.

However, the paper requires a major revision in the following areas:

  • Introduction: Additional references should be included to highlight the role of PIVKA-II as a biomarker in various neoplasms, such as pancreatic ductal adenocarcinoma (PDAC), emphasizing its emerging relevance beyond HCC.

Reply

According to reviewer’s comments, we added new additional reference (see reference 7) related to the relationship between PDAC and PIVKA-II.

Page 16, Line 25

  Farina, A.; Tartaglione, S.; Preziosi, A.; Mancini, P.; Angeloni, A.;

  Anastasi, E. PANC-1 cell line as an experimental model for

  characterizing PIVKA-II production, distribution, and molecular

  mechanisms leading to protein release in PDAC. Int. J. Mol. Sci. 2024,

  25, 3498. https://doi.org10.3390/ijms25063498

  • Hypothesis: The proposed compensatory role of prothrombin precursor production is based on indirect evidence. Therefore, data on serum precursor levels should be incorporated to support this hypothesis.

Reply

In Japan, it is not easy to measure prothrombin precursors and our facility does not have the equipment to measure them. Therefore, we could not measure serum levels of prothrombin precursors, and thus the increased formation of prothrombin precursors was not serologically proven, and it was indirectly proven according to the trends in prothrombin levels.

  • Results: Histological analyses confirming PIVKA-II production in tissues should be included.

Reply

I am sorry, but we could not perform the liver biopsy for confirmation of PIVKA-II production in tissues, because this study was conducted among outpatients. Therefore, its histological data could not be obtained.

  • Discussion: Some sections are redundant and should be streamlined. Additionally, a comparison with findings from other studies would be valuable, highlighting similarities, differences, and potential clinical implications.

Reply

According to reviewer’s comments, we deleted unnecessary sections or similar sentence as much as possible.

Page 7. Line 29

    Based on our results, we speculated on the possible relationship

   between each coagulation factor in an accelerated coagulation

   -fibrinolysis system in hepatic hemangiomas (Figure 1). A previous

   study reported that fibrinogen levels were correlated with platelet

   counts [27], and our results also demonstrated that fibrinogen levels

   were significantly correlated with platelet counts. Low platelet counts

   and fibrinogen levels were considered to be the result of intravascular

   coagulation in hemangiomas [28]. Almost all patients with elevated

   levels of D-dimer and FDP exhibited increased levels of TAT.

   Furthermore, a significant correlation was observed between each of

   five coagulation factors (platelet, fibrinogen, TAT, D-dimer, and FDP)

   except prothrombin. Therefore, it is assumed that prothrombin behaves

   differently from these five other coagulation factors in the process of

   the coagulation-fibrinolysis, resulting from compensatory increased

   synthesis of prothrombin in hepatic cells.

   Page 8, Line 10

   (Our previous studies showed that PIVKA-II levels were significantly

   elevated in patients with larger hemangiomas and coagulation disorders

   [7-9], and → delete)

    The present results relating to the change in PIVKA-II levels by tumor

   growth demonstrated an elevation of PIVKA-II levels in patients with

   increases in tumor size and abnormal coagulation factors, no change in

   patients with stable lesions, and a decrease in patients with decreases in

   tumor size and abnormal coagulation factors. Furthermore, PIVKA-II

   levels were significantly correlated with tumor size and all coagulation

   factors except prothrombin. From the above results, PIVKA-II levels

   were closely associated with changes in hemangioma size and

   abnormal coagulation factors.

  • Conclusion: The conclusion should be more impactful, emphasizing the clinical significance and implications of the findings.

Reply

According to reviewer’s comments, conclusion was changed to the following in order to emphasize the clinical implications.

   Page 11, Line 23

   Conclusions

     Present study is the first to demonstrate the correlation of hepatic

   hemangiomas with serum PIVKA-II levels. Hemangiomas were

   significantly associated with elevated serum PIVKA-II levels, which

   was attributed to increased production of prothrombin precursors

   caused by the accelerated coagulation-fibrinolysis within

   hemangiomas. Although further studies are required to evaluate the

   exact relationship between hepatic hemangiomas and PIVKA-II

   levels, we recommend that elevated serum PIVKA-II levels in patients

   with hemangiomas should be interpreted with close attention to

   differentiating them from HCC.

  • Limitations: Consider adding a clearer section outlining the study’s limitations.

Reply

According to reviewer’s comments, we eleted some parts and added a clearer section.

Page 9, Line 15

     The present study has several limitations. First, serum levels of

   prothrombin precursors were not measured, and thus the increased

   formation of prothrombin precursors was not serologically proven, and

   it has only been indirectly proven according to the trends in

   prothrombin levels. Second, the levels of prothrombin, vitamin K, and

   PIVKA-II in liver tissues were not measured, and our views were

   established based on the serological levels; therefore, measurements

   and results at the histological level are needed to validate the role of

   PIVKA-II in hemangiomas. Third, the length of follow-up and the

   number of patients, especially those with giant hemangiomas, were

   limited; therefore, a longer follow-up period and a larger patient cohort

   with giant hemangiomas are needed to evaluate the elevation of serum

   PIVKA-II levels. Therefore, further studies are required to evaluate the

   exact relationship between hemangiomas and the elevation of serum

   PIVKA-II levels.

Round 2

Reviewer 1 Report

Comments and Suggestions for Authors

Most of the issues raised by the reviewers have been addressed meaningfully in the revised version of the manuscript. It may now be accepted for publication.

Author Response

Thank you so much for your peer review. 

Reviewer 2 Report

Comments and Suggestions for Authors

Accept in the present form

Comments on the Quality of English Language

Accept in the present form

Author Response

Thank you so much for your peer review. I am looking for publication.